# Liver Damage Associated with SARS-CoV-2 Infection—Myth or Reality?

**DOI:** 10.3390/jpm13020349

**Published:** 2023-02-17

**Authors:** Mihaela Cătălina Luca, Isabela Ioana Loghin, Ioana Florina Mihai, Radu Popa, Andrei Vâţă, Carmen Manciuc

**Affiliations:** 1Department of Infectious Diseases, “Grigore T. Popa” University of Medicine and Pharmacy, 700115 Iasi, Romania; 2Department of Infectious Diseases, “St. Parascheva” Clinical Hospital of Infectious Diseases, 700116 Iasi, Romania; 3Emergency County Hospital, 600114 Bacau, Romania; 4Vascular Surgery Department, “St. Spiridon” Emergency Clinical Hospital, 700111 Iasi, Romania

**Keywords:** liver damage, SARS-CoV-2, coronavirus, serum aminotransferases

## Abstract

(1) Introduction: While the primary impairment in COVID-19 is pulmonary, the ubiquitous distribution of angiotensin-converting enzyme 2 (ACE2) demonstrates the possible presence of systemic disease with involvement of the heart, kidneys, liver and other organs. (2) Methods: We retrospectively studied the observation sheets of patients diagnosed with SARS-CoV-2 infection hospitalized in the “Sf. Parascheva” Clinical Hospital of Infectious Diseases from Iasi for a period of 3 months. The aim of the study was to identify the frequency of liver injury due to SARS-CoV-2 infection among patients and its impact on the course of the disease. (3) Results: Out of the total number of hospitalized cases (1552), 207 (13.34%) were the subjects of our analysis. The severe form of SARS-CoV-2 infection predominated (108 cases; 52.17%) and in terms of liver damage, in all cases increased transaminase levels predominated and were determined to be secondary to the viral infection. We divided the lot into two groups, A (23 cases; 23.19%) and B (159 cases; 76.81%), depending on the time of onset of liver dysfunction, either at the time of hospitalization or during hospitalization. The evolution of liver dysfunction was predominant in most cases, with an average time of onset at 12.4 days of hospitalization. Death occurred in 50 cases. (4) Conclusions: This study revealed that high AST and ALT at hospital admission was associated with a high mortality risk in COVID-19 patients. Therefore, abnormal liver test results can be a significant prognostic indicator of outcomes in COVID-19 patients.

## 1. Introduction

Severe acute respiratory syndrome coronavirus 2 (SARS-CoV-2) is the third strain of the Coronaviridae family responsible for causing a pandemic. Coronavirus disease 2019 (COVID-19), the respiratory disease responsible for the ongoing COVID-19 pandemic is caused by the SARS-CoV-2 virus. Its appearance in December 2019 in Wuhan, China and its rapid spread in over 216 countries led to an unprecedented global crisis with repercussions on the medical and economic system, but also on the quality of life of individuals [1]. The similarity in the clinical manifestations of SARS-CoV-2 infection with other diseases, the unpredictability of the clinical and paraclinical evolution, the uncertainties regarding side effects and the treatment of the infection, and the impact of the self-isolation/isolation/quarantine regime were real challenges [2].

The experience of the current pandemic has shown us that SARS-CoV-2 can affect various organs and systems such as the cardiovascular, respiratory, neurological, digestive, metabolic systems, and more. Older adults and patients with a significant personal history of medical pathology (i.e., metabolic syndrome, cardiovascular disease, neurological disease, oncological, or liver disease) may be at an increased risk for the severe COVID-19 form [3]. Oxygen saturation (SpO2) < 94% in room temperature air at sea level, the ratio of arterial oxygen partial pressure (PaO2 in mmHg) to fractional inspired oxygen (PaO2/FiO2) < 300 mm Hg, a respiratory rate > 30 breaths/min, or lung infiltrates > 50% defines a severe SARS-CoV-2 illness. These patients present with an evolution towards rapid clinical deterioration [4].

Liver disease is among the main extrapulmonary manifestations. Liver damage associated with SARS-CoV-2 infection is defined as any liver damage occurring during the disease course and treatment of COVID-19 in patients with or without pre-existing liver disease. According to the statistical data up to now, in one out of five patients diagnosed with SARS-CoV-2 infection, we also find liver damage [5]. In such cases, blood test results highlight elevated liver enzymes such as alanine aminotransferase (ALT) and aspartate aminotransferase (AST). Also, alkaline phosphatase (ALP) and gamma-glutamyl transferase (GGT) are enzymes that suggest liver damage. Some studies have shown that people with pre-existing liver disease (chronic liver disease, cirrhosis, or related complications) who have been diagnosed with COVID-19 have a higher risk of death than people without pre-existing liver disease [6,7].

Mechanically, SARS-CoV-2 infection involves the binding of the viral spike protein to angiotensin-converting enzyme 2 (ACE2) as an entry receptor on the cell surface. This binding facilitates the penetration of the virus into the cell, viral replication, and intercellular transmission [8]. Due to the ubiquitous distribution of ACE2, SARS-CoV-2 causes systemic disease, with possible involvement of the heart, kidneys, liver, or other organs, causing changes in circulating lymphocytes and the immune system. Liver ACE2 receptors are expressed mainly in cholangiocytes (59.7% of cells), minimally expressed in hepatocytes (2.6% of cells), and absent in Kupffer cells. Thus, it is confirmed that SARS-CoV-2 infection affects liver function by direct cytotoxicity due to the continuous replication of the virus in the previously mentioned cell populations [9,10].

The inflammatory immune response may be responsible for hepatic impairment in COVID-19. This is demonstrated by the presence of elevated values of inflammatory markers (e.g., C-reactive protein (CRP), interleukin-6 (IL-6), interleukin-2 (IL-2), lactate dehydrogenase (LDH), ferritin, D-dimers) suggesting a direct link between the presence of cytokine storm syndrome and the severity of the disease [11].

In addition, there are other factors that contribute to impaired liver function, including medication used to treat SARS-CoV-2 infection (e.g., umifenovir, remdesivir, tocilizumab, lopinavir/ritonavir) or the treatment of associated bacterial infections (antibiotics, antipyretics, steroids), pre-existing liver disease, congestion liver, or ischemic hypoxic lesions [12].

Our goal was to highlight the effects of SARS-CoV-2 infection on the liver. For this purpose, we conducted a study on a group of patients diagnosed with SARS-CoV-2 infection. Finally, we used data and developing theories from the literature and corroborated them with the data obtained by our research.

## 2. Materials and Methods

### 2.1. Studied Patients

We conducted a retrospective clinical study focusing on hospital-based medical records of patients from “Sf. Parascheva” Clinic Hospital of Infectious Diseases from Iasi, having the objective of evaluating patients diagnosed with SARS-CoV-2 infection and associated liver injuries. Patients included were hospitalized between the 1 October 2021 and 31 December 2021.

### 2.2. Inclusion and Exclusion Criteria

All hospitalized patients over 18 years of age who tested positive for SARS-CoV-2 infection and had transaminase levels more than three times the normal value in the absence of pre-existing liver damage were included in the study. Diagnosis of SARS-CoV-2 infection was based on real-time reverse transcriptase-polymerase chain reactions (RT-PCR) performed on either nasopharyngeal swabs (NPS) or oropharyngeal swabs (OPS). Patients who tested positive for infection but had no evidence of liver damage with normal or quasi-abnormal transaminases or had a known pre-existing liver disease were excluded from this study.

### 2.3. Data Collection

We collected from each patient’s medical database: demographic data (age, sex), medical history, clinical features, blood tests, in-hospital treatment and outpatient outcomes (severe COVID-19, intensive care unit (ICU), mechanical ventilation, and death). Liver test abnormalities were defined according to reference laboratory standards: AST (aspartate aminotransferase) > 37 U/L, ALT (alanine aminotransferase) > 40 U/L, ALP (serum alkaline phosphatase) outside the range of 98–279 U/L, TBIL (total-value bilirubin) > 1 mg/dL, and gamma-glutamyl transferase (GGT) > 49 U/L. All blood tests were performed by the hospital’s central laboratory. RT-PCR tests were performed either by the hospital’s molecular biology laboratory or by other accredited laboratories, on either nasopharyngeal swabs (NPS) or oropharyngeal swabs (OPS).

Only patients whose transaminase values were increased more than three times were included in the study because we wanted to do a statistically significant analysis of liver damage due to SARS-CoV-2 infection. The chosen period, namely October to December 2021, was representative due to the predominance of the Delta strain in Romania. This was the period with the most admissions in the entire pandemic, the most serious cases, and an increase in the number of deaths.

### 2.4. Statistical Analysis

Correlations between demographic parameters, clinical data, and outcomes were performed using Pearson test in XLSTAT version 2019 software. Kendall’s Τau correlation coefficients were calculated [13]. Statistical analysis was performed using Statistical Software for Excel (XLSTAT) version 2019.

## 3. Results

During the abovementioned period in the “St. Parascheva” Clinical Hospital of Infectious Diseases from Iasi, 1552 patients confirmed to have SARS-CoV-2 infection by reverse transcriptase real-time quantitative polymerase chain reaction (RT-PCR SARS-CoV-2 RNA) were hospitalized. All patients were tested for SARS-CoV-2 infection before or on the day of hospitalization. From the total number of hospitalized cases in the mentioned period, 500 cases with liver damage were observed. After excluding pre-existing liver pathology (chronic hepatitis, liver cirrhosis, portal hypertension syndrome, ascites, jaundice, liver failure, and biliary colic), 207 patients demonstrated transaminase levels more than three times the normal value in the absence of pre-existing liver damage. Our study includes an analysis of the 207 cases mentioned above.

### 3.1. Patients’ Characteristics

The age of the patients varied between 18 and 94 years, with an average age of 61 years. The most affected patients were those in the age group of 60–79 years (102 patients, 49.27%) (Figure 1).

The total group of patients included in the study (207 in total) was divided into two groups. We called them group A and group B. Group A (48 patients, 23.19%) was represented by the category of patients with whom an increase in transaminase values was observed since the beginning of hospitalization; we considered an increase of more than three times of the normal value statistically significant. Group A also included patients where, after hospitalization and the initiation of treatment, the values decreased to the normal value or demonstrated values close to normal.

Group B (159 patients, 76.81% of the total) included patients who had normal values of transaminases or values close to normal at the time of hospitalization and those who had more than three times increase in the values normal for AST and ALT during the hospitalization and the initiation of the treatment.

The gender distribution and residence of patients showed a slightly higher share of male patients (52.65% vs. 47.34% female patients, 109 males vs. 98 female, *p* < 0.05) and those living in urban areas (56.52% vs. 43.48% in rural areas, *p* < 0.05) (Table 1).

The detailed anamnesis of the patients highlighted the fact that the average time from symptom onset to hospital admission was 7.7 days. Eighteen patients (8.17%) had a complete COVID-19 vaccination scheme at hospitalization, 6 patients (2.90%) had an incomplete vaccination scheme, and the remaining 183 patients (88.40%) did not have specific anti-SARS-CoV-2 prophylaxis. A total of 39.47% (61/207) patients had no comorbidities. More than half of them had pre-existing pathologies. Excluding liver damage, patients in the study had cardiovascular disease (56 patients, 27.37%), kidney disease (47 patients, 22.70%), diabetes (27 patients, 13.04%), and neurological impairment (17 patients, 8.21%).

According to the definitions developed by the World Health Organization, the classification of COVID-19 in terms of clinical form is mild, moderate, or severe depending on the degree of respiratory function [14]. Regarding the patients in the study group, most developed a severe form of SARS-CoV-2 infection (108 patients, 52.17%). Four (1.93%) patients developed a mild form and 95 (48.89%) developed a moderate form.

### 3.2. Laboratory Findings

Hepatic impairment was highlighted primarily by increased transaminases (AST, ALT). All cases with elevated transaminases were determined to be secondary to SARS-CoV-2 infection. In 48 cases (23.19%), transaminase abnormalities were highlighted since the beginning of hospitalization and then subsequently normalized (group A) (Figure 2). In 159 cases (76.81%) an increase in liver enzymes was observed after 6 days of hospitalization, with a decrease towards the normal level (group B) (Figure 3).

In the graphs above, the changes in laboratory values regarding liver enzymes AST and ALT are exemplified in the case of all patients. In group A there were 5 cases with AST and ALT values over 250 U/L and in group B, in 15 cases an increase of over 250 U/L was observed during hospitalization. In the other cases, a transient increase of up to 250 U/L of transaminases was observed.

Correlated with transaminase values, we also evaluated the level of total bilirubin, with a predilection for indirect bilirubin, gamma-glutamate dehydrogenase, and alkaline phosphatase. We evaluated platelet numbers, total proteins, and prothrombin values in all patients in the study. An analysis of laboratory data showed patients with hepatic impairment had elevated indirect bilirubin (19 cases, 9.17%) and increases in gamma glutamate dehydrogenase (GGT) (72 cases, 34.78%), thrombocytopenia (46 cases, 22.21%), and hypoalbuminemia (18 cases, 8.70%).

### 3.3. Treatment

In hospital the treatment used for SARS-CoV-2 infection was in accordance with the legislation in force at that time. The following medications were administered to patients: antibiotics such as Ceftriaxone (137 cases, 66.18%), Imipenem/cilastin (62 cases, 29.95%), and Linezolid (50 cases, 24.15%); antivirals such as Favipiravir (109 cases, 52.65%) and Remdesivir (46 cases, 22.22%); anti-inflammatory and immunomodulatory medications such as Anakinra (82 cases, 39.61%) and Tocilizumab (16 cases, 7.72%) (Figure 4); and vitamins according to symptoms. Hepatoprotectives and amino acids have been used for liver damage due to their detoxifying effect on the liver.

Prior to hospitalization all patients reported self-administration of various treatments, with most of them being symptomatic. A percentage of 18.84% (39/207) patients combined antibiotic/antiviral treatments at home (Figure 5).

### 3.4. Evolution

The average time from hospital admission to discharge/interhospital transfer/death was 12.4 days. The evolution was favorable in more than half of the cases. Out of the total number of patients, 76 (36.71%) required hospitalization in the intensive care unit. In 19 cases (9.17%) patients were transferred to specialized clinics depending on the associated comorbidities and in 50 cases (24.15%) death occurred (32 cases due to multiple organ dysfunction and 18 cases due to severe respiratory failure).

In group A, 8 deaths were reported, two of which were associated with severe liver failure since admission (case 1 of a 65-year-old male patient with a severe form of COVID-19 who had AST 10 618 U/L and ALT 4260 U/L and case 2 of an 85-year-old woman with a severe form of COVID-19 who had an AST of 5700 U/L and an ALT of 2115 U/L). In both cases, the evolution was slow and worsening, with the need to monitor and continue treatment in the intensive care unit for more than 20 days.

Group B recorded a statistically higher percentage of deaths (84%). There were 4 cases in which the patients had associated liver failure. Only in one case did the death occur due to worsening liver function—the case of the 37-year-old patient with a severe form of COVID-19 who had an ALT of 11,255 U/L and an AST of 16,455 U/L.

## 4. Discussion

The new coronavirus began to be talked about at the end of 2019 after five cases of a respiratory disease of unknown origin were identified, with patients showing symptoms similar to viral pneumonia [15]. From then until now (6 February 2023), after almost 3 years of the COVID-19 pandemic, more than 675 million cases of SARS-CoV-2 infection have been identified globally, with more than 6 million deaths [16]. A new reason for concern for the entire medical system was the appearance of new variant strains that presented several mutations of the virus. They were named Alpha (British strain—B. 1. 1. 7), Beta (South Africa—B. 1. 351), Gamma (Brazilian strain—P. 1), Delta (Indian strain—B. 1. 617. 2), and Omicron (South Africa—B. 1. 1. 529) [17,18]. Variant B1617.2 (Delta), or the Indian strain, is considered the most adapted of the four viral variants of concern, with a high degree of infectivity twice that of the original strain and about 60% greater than the Alpha variant, known as the British strain. Detected variants of coronavirus have caused fears at the global level so a great emphasis has been placed on vaccination campaigns [19].

Although a pathology of the respiratory system which was unknown to the whole world at the beginning of 2020, the SARS-CoV-2 infection proved to be multisystemic, affecting several organs and systems. From respiratory, digestive, and skin system damage to liver, metabolic, and cardiovascular damage, the clinical picture of SARS-CoV-2 infection proved to be diverse [15].

Liver disease has been reported since the beginning of the discovery of the novel coronavirus, frequently being described as a common manifestation, although its clinical significance was not fully understood, especially for patients with underlying chronic liver disease (CLD) [20]. Both in the case of patients with pre-existing liver damage as well as in the case of patients with de novo liver damage, liver damage in hospitalized patients is mainly manifested by changes in biochemical markers in the liver [21].

Liver damage presenting with increased liver enzymes and lactate dehydrogenase was first reported in a study of 99 patients in early 2020 [22]. Of the total, 43 (43.43%) had varying degrees of liver damage and impaired liver function with ALT or AST above the normal range. In one case, the value of transaminases indicated severe acute impairment of liver function, with serum ALT and AST levels increased up to 7590 and 1445 U/L, respectively, in a patient with the severe form of COVID-19 [23]. A review article reported abnormal levels of ALT and AST during COVID-19 ranging from 14% to 53% [24].

Another study of nearly 1100 patients in China observed that elevated serum AST levels were found in approximately 18% of patients with moderate or mild COVID-19 disease and in approximately 56% of patients with the severe COVID-19 form. Moreover, in that study, elevated serum ALT levels were observed in nearly 20% of patients with mild or moderate COVID-19 disease and in approximately 28% of patients with severe COVID-19 disease [25]. In cases of COVID-19 resulting in death, the incidence of liver injury might reach as high as 58.06 and 78% [26].

Liver test abnormalities were defined as the elevation of the following liver enzymes in serum: ALT > 40 U/L, AST > 37 U/L, GGT > 49 U/L, ALP outside the range of 98–279 U/L, and TBIL > 1 mg/dL [27]. In our study, it was highlighted that 500 patients had liver damage, with 293 cases having pre-existing liver damage. Most of the cases did not have a significant deterioration of liver function and, for this reason, they were excluded from the study. We focused on the 207 cases, showing that liver involvement during COVID-19 infection with ALT or AST three times above the normal value may affect less than a quarter of patients admitted to the clinic (207 cases out of a total of 1552, 13.34%). In group A there were 5 cases with AST and ALT values over 250 U/L and in group B, in 15 cases an increase of over 250 U/L was observed during hospitalization. In the other cases, a transient increase of up to 250 U/L in transaminase values was observed.

According to the statistical data published so far, the increase in ALT was observed between day 4 and day 17 of hospitalization, at an average of 7.3 ± 3.0 days in severe forms compared to 10.7 ± 4.1 days in mild forms (*p* = 0.048) [28]. The batch studied by us was divided into two groups based on the moment of occurrence of de novo liver damage. Thus, we called them group A and group B. Group A (48 patients, 23.19%) was represented by the category of patients with whom increased values of liver enzymes were present from the first day of hospitalization; we considered an increase of more than three times the normal value statistically significant. Group A also included patients with whom, after hospitalization and the initiation of treatment, the values decreased to the normal value or demonstrated values close to normal.

Group B (159 patients, 76.81% of the total) included patients who had normal values of transaminases or values close to normal at the time of hospitalization and those who had more than three times increase in the values normal for AST and ALT during the hospitalization and the initiation of the treatment. In these cases, the increase in transaminase values was highlighted in the first week of hospitalization, after the 6th day.

Although serum transaminases may have been elevated before COVID-19, results from clinical reports and autopsy studies suggest that liver dysfunction may be an expression of a more severe course of the disease and that an increase in transaminases alone is likely to be indirect expression of systemic inflammation [24].

In another study of 417 patients with COVID-19, abnormal liver tests (i.e., AST, ALT, total bilirubin, GGT) were present in 76.3% and 21.5% developed liver damage during hospitalization, especially within the first two weeks after admission.

Studies have shown that patients with abnormal liver tests have a higher risk of progressing to severe disease and liver impairment is closely linked to mortality in patients with COVID-19 [21]. In our study, the severe form of COVID-19 was found in 108 cases (52.17%), accounting for more than half of all cases. Also, correlating the value of transaminases with the severity of COVID-19, we observed that in severe forms of the disease, the transaminases had a value 10 times higher than the normal one, in the case of ALT in 27 patients (9.9%) and in the case of AST in 15 patients (5.4%). The highest value was recorded in a 37-year-old patient with a severe form of COVID-19 (ALT 11,255 U/L, AST 16,455 U/L) from group B who presented at the hospital on the third day after the onset of symptoms. On the day of admission, the values of the laboratory analysis showed the following values: ALT 159 U/L, AST 109 U/L, GGT 334 U/L, ALP 437 U/L, and BT 2.68 mg/dL. The evolution in the case of this patient was unfavorable. Being a severe form of COVID-19, it required continuous oxygen therapy, supervision, and treatment in the intensive care unit, exhibiting a rapid deterioration of organs and systems. It was the first case in the study in which liver failure was one of the causes of death.

Regarding the evolution of the cases, improvement was noticed in more than half of the cases, with an average time from hospital admission to discharge/interhospital transfer/death of 12.4 days. Out of the total number of patients, 76 (36.71%) required hospitalization in the intensive care unit. In 19 cases (9.17%) patients were transferred to specialized clinics depending on the associated comorbidities and in 50 cases (24.15%) death occurred (32 cases due to multiple organ dysfunction and 18 cases due to severe respiratory failure).

In group A, 8 deaths were reported, two of which were associated with severe liver failure present since admission (case 1 of a 65-year-old male patient with the severe form of COVID-19 who had AST 10,618 U/L and ALT 4260 U/L and case 2 of an 85-year-old woman with the severe form of COVID-19 who had AST 5700 U/L and ALT 2115 U/L). In both cases, the evolution was slow and worsening, with the need for monitoring and continuing treatment in the intensive care unit for more than 20 days.

Group B recorded a statistically higher number of deaths. There were 4 cases in which the patients had associated liver failure and only in one case did the death occur due to worsening liver function, that being the case of the 37-year-old patient with the severe form of COVID-19 who had ALT 11,255 U/L and AST 16,455 U/L.

The prevalence of abnormal liver tests may also be due to the medications used during hospitalization. In a study conducted at the Third People’s Hospital of Shenzhen from China [29], among a total of 170 patients, 84% used lopinavir/ritonavir during hospitalization, which was reported to cause liver damage and affect liver tests. In our study, the medications administered to the patients were: antiviral treatment: Favipiravir (109 cases—52.65%) and Remdesivir (46 cases—22.22%); anti-inflammatory and immunomodulatory medication: Anakinra (82 cases—39.61%) and Tocilizumab (16 cases—7.72%); along with vitamins, hepatoprotectives, and amino acids according to symptoms, the latter of which is used for liver damage due to its detoxifying effect on the liver.

In addition to the increase in transaminase values, we were able to observe in our study increases in indirect bilirubin and gamma-glutamate dehydrogenase, with unchanged levels of alkaline phosphatase. Among patients with pre-existing liver pathology, it was also possible to observe a decrease in platelet values and a decrease in total proteins, especially albumin.

Hepatic impairment in mild cases of COVID-19 is often transient and may return to normal without special treatment. However, when severe liver damage occurs, liver protection treatment should be given to patients [30].

Therefore, if we refer to the causes of liver damage in our study, we refer to pre-existing liver damage, the direct expression of viral infection in the liver, and the toxic liver action of the medication used in treatment. Also, if we refer to the hepatic effects of SARS-CoV-2 infection on the patients in the study, there were increases in liver enzymes, increased total bilirubin, increases in glutamate dehydrogenase, decreased platelets, decreased bilirubin, and unchanged alkaline phosphatase.

## 5. Conclusions

Liver damage in the case of patients diagnosed with SARS-CoV-2 infection is one of the main manifestations of the infection, especially in hospitalized patients, and its presence has been associated with an increased risk of complications, including death. The cause of liver damage is most likely viral infection of bile duct cells or functional impairment caused by the use of antiviral drugs.

The present study revealed that high AST and ALT at hospital admission was associated with severe forms of COVID-19 and a high mortality risk in COVID-19 patients. Therefore, abnormal liver test results can be a significant prognostic indicator of outcomes in COVID-19 patients.

In COVID-19 elevated liver enzymes are usually mild and generally recover with hepatoprotective treatment or regimens and rest. However, the increased values of transaminases in the context of severe forms of COVID-19 or as a consequence of antiviral therapies with effects on the liver should not be neglected. It is therefore necessary to distinguish in clinical practice if the onset of abnormal liver function occurs at diagnosis or during treatment.

The mechanism of liver damage in patients with SARS-CoV-2 infection is multifactorial and requires additional research to improve the therapeutic approach in this category of patients.

Even if the pandemic is officially over, cases of infection are still being diagnosed all over the world so future research should continue. Deciphering virus–host interactions will certainly give us more clues about the detailed molecular mechanisms of liver failure following SARS-CoV-2 infection.

## Figures and Tables

**Figure 1 jpm-13-00349-f001:**
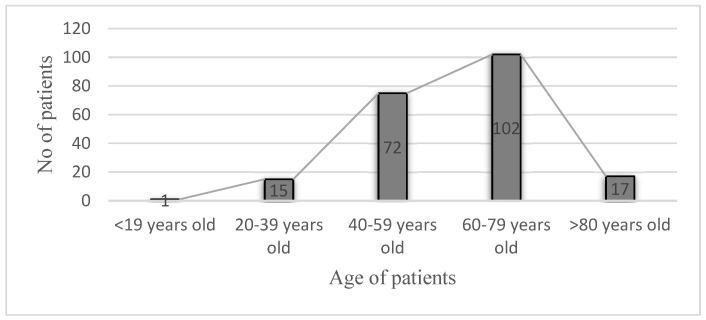
Number of patients according to age group.

**Figure 2 jpm-13-00349-f002:**
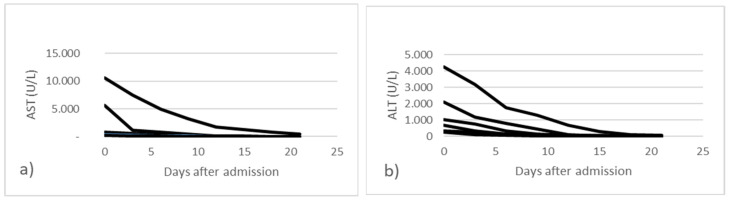
Dynamic changes in transaminase levels in all patients from group A; (**a**) aspartate aminotransferase and (**b**) alanine transaminase.

**Figure 3 jpm-13-00349-f003:**
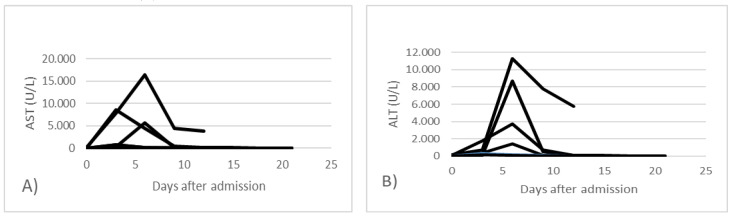
Dynamic changes in transaminase levels in all patients from group B; (**A**) aspartate aminotransferase and (**B**) alanine transaminase.

**Figure 4 jpm-13-00349-f004:**
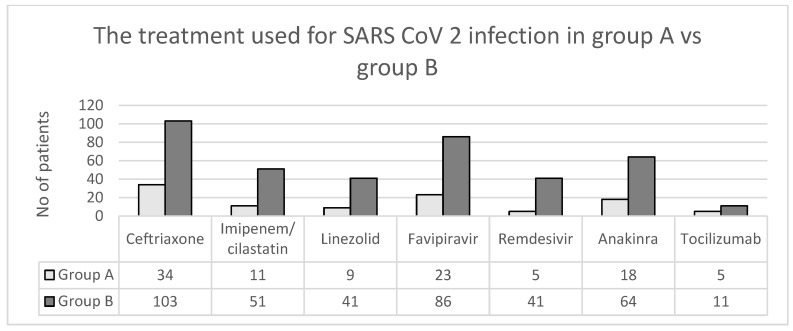
The treatment used for SARS-CoV-2 infection in group A versus group B.

**Figure 5 jpm-13-00349-f005:**
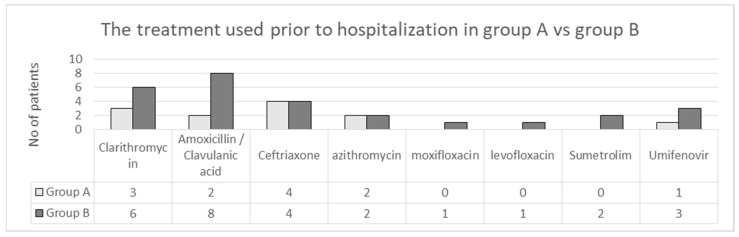
The treatment used prior to hospitalization in group A versus group B.

**Table 1 jpm-13-00349-t001:** Demographic characteristics of the patients enrolled in our study.

Number of Patients Enrolled	207		100%		*p*
Gender	Male	Female	Male	Female	*p* < 0.05
	109	98	52.65%	47.34%	
Residence	Urban	Rural	Urban	Rural	*p* < 0.05
	117	90	56.52%	43.38%	

## Data Availability

All data generated or analyzed during this study are included in this published article.

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
