# Peer review of "Liver Damage Associated with SARS-CoV-2 Infection—Myth or Reality?"

_jpm, 2023, doi:10.3390/jpm13020349_

Round 1

Reviewer 1 Report

The paper is interesting since it contains statistical clinical data. However, the Results sometimes are not easy to follow, and the Conclusions can be improved. 

1. Why only AST (not ALT and AST) are indicated in the Abstract and in the Conclusions? Both enzymes elevations seem significant.

2. Please indicate in Abstract the total number of hospitalized cases (1552) before the indication of 207 cases taken for the analysis.

3. Figure 1: there are 210 patients indicated in total. In the text the authors indicate they have analyzed 207 patients.

4. In section 3.2. the two group of patients (A and B) should be described more clearly. Please describe first Group A and then Group B. 

Line140:  "...with the decrease towards the normal limit (Group B)" What have the authors meant? Do people from Group B achieved the normal limit during the hospitalization? As seen from Figure 2-3, it is Group A, where patients decreased their ALT/AST levels during hospitalization, not Group B.

5. "Figure 2-3" is not generally accepted. Please  make one figure with two panels (a, b).

6. What are the units in the Y-axis in Figure 2-3? The "Transaminase values" indicated in Graphs seems to be very high for one person. Is it a collective (summarized) value for a group? (how many persons collects each bar?)

7. There are a lot of values indicated in the manuscript (number of patients; number of patients as a percentage) which are very difficult to follow. Could the authors make one or two Tables to summarize the main Results they obtain? E.g. how many persons have been hospitalized; how many of them possesed seriously elevated ALT/AST values; how many were in Group A/B; how many from each group were recovered, transfered to another hospital/hospital department, how many died from each group.

Is it known exactly about the patients who died (50 deaths): are they all belong to Group B?

8. Line 85: "...achieved by chain reaction of reverse transcriptase-polymerase" is not correct.

Reviewer 2 Report

Thank you for giving me this opportunity to review this article. This article needs major revisions and improvement in the analysis and results.

It is not clear why only transaminase levels more than 3 times the normal is included to achieve this objective and also only 3 months data is included. The statistical analysis mentioned and the analysis done/presented in the results section is not matching. The title/introduction not matched with the results presented and not concluded in the conclusion.

Round 2

Reviewer 1 Report

The authors answered many of my concerns.

There are some points remained to be corrected/clarified:

1. Line 28: please change "AST and AST" for "AST and ALT";

2. Line 33: must read "1. Introduction";

3. Line 110: "in our country" would be better to replace by "in Romania";

4. Line 123: "500" should be replaced by "1552";

5. Line 166: "limit" should be replaced by "level";

6. Lines 169-170 and 172-173: There should be indicated in legends to Figure 2 and Figure 3 which cases are shown as curves in those figures: the most serious? typical? There are just few curves represented from a large sampling of cases. Why? Muct be also explained/discussed in the text (in the Results section);

7. Line 176 (an old version of Figure 2,3) must be removed;

8. Lines 201-207 (Evolution section): is it possible to say anything about good or poor prognosis regarding patients from Group A and Group B? Did Group A demonstrate more percent of good prognosis campared to Group B? E.g. do all died 50 patients belong to Group B or to different Groups (A and B)? Do the authors have such statistical data?

9. Line 238: "...than the normal one. In the case of ALT..." the dot should be removed, and there should be one sentence instead of two sentences made;

10. Line 239-241:  "The highest value was recorded in a 73-year-old patient with a severe form of COVID-19 (ALT 1210 U/L, AST 2400 U/L)."

Are these really the biggest ALT and AST values the authors recordered?

There are much higher values indicated in Figures 2 and 3. And the Authors' Response to Revision 1 (point 5) says that "Example – a severe case had AST value 11.255 U/L and ALT 16.500 U/L". These data should be consistent and properly discussed.

11. Line 266: Please replace "live" for "liver".
